# The Effect of the Gut Microbiota on Systemic and Anti-Tumor Immunity and Response to Systemic Therapy against Cancer

**DOI:** 10.3390/cancers14153563

**Published:** 2022-07-22

**Authors:** Azin Aghamajidi, Saman Maleki Vareki

**Affiliations:** 1Department of Immunology, School of Medicine, Iran University of Medical Sciences, Tehran 1449614535, Iran; azinaghamajidi@gmail.com; 2Department of Pathology and Laboratory Medicine, Western University, London, ON N6A 3K7, Canada; 3London Regional Cancer Program, Lawson Health Research Institute, London, ON N6A 5W9, Canada; 4Department of Oncology, Western University, London, ON N6A 3K7, Canada; 5Department of Medical Biophysics, Western University, London, ON N6A 3K7, Canada

**Keywords:** immunotherapy, immune checkpoint inhibitors, microbiota, cancer

## Abstract

**Simple Summary:**

The gut microbiome affects the development of systemic immune response and it can also impact response to systemic treatments such as immunotherapy and chemotherapy. This article provides and in-depth overview of various mechanisms that the gut microbiome interacts with the immune system, cancer, and how it affects anti-tumor immunity and response to immunotherapy.

**Abstract:**

Gut microbiota can have opposing functions from pro-tumorigenic to anti-tumorigenic effects. Increasing preclinical and clinical evidence suggests that the intestinal microbiota affects cancer patients’ response to immune checkpoint inhibitors (ICIs) immunotherapy, such as anti-programmed cell death protein 1 (PD-1) and its ligand (PD-L1) and anti-cytotoxic T lymphocyte-associated protein 4 (CTLA-4). Microbiota-induced inflammation possibly contributes to tumor growth and cancer development. Microbiota-derived metabolites can also be converted to carcinogenic agents related to genetic mutations and DNA damage in organs such as the colon. However, other attributes of microbiota, such as greater diversity and specific bacterial species and their metabolites, are linked to better clinical outcomes and potentially improved anti-tumor immunity. In addition, the intratumoral microbial composition strongly affects T-cell-mediated cytotoxicity and anti-tumor immune surveillance, adding more complexity to the cancer-microbiome-immune axis. Despite the emerging clinical evidence for the activity of the gut microbiota in immuno-oncology, the fundamental mechanisms of such activity are not well understood. This review provides an overview of underlying mechanisms by which the gut microbiota and its metabolites enhance or suppress anti-tumor immune responses. Understanding such mechanisms allows for better design of microbiome-specific treatment strategies to improve the clinical outcome in cancer patients undergoing systemic therapy.

## 1. Introduction

Cancer is a genetic- and environmentally- influenced disease recognized as a global health problem given its relatively high morbidity, mortality, and economic cost. The process of carcinogenesis often includes genetic and epigenetic alterations, leading to chromosomal aberration and uncontrolled cell division. Different exogenous and endogenous determinants, such as smoking, alcohol consumption, obesity, and genetic predispositions, are primary contributors to cancer development [1]. As an evolutionary protective mechanism, the immune system has evolved to identify and eliminate neoplastic cells to prevent cancer from growing and spreading to other organs. Tumor immune surveillance includes various immunological effector mechanisms involving innate and adaptive immunity [2]. However, the long-lasting chronic stimulation of immune cells by tumor antigens and uncontrolled inflammation associated with carcinogenesis can eventually impair anti-tumor immunity and promote tumor progression. Therefore, the ability of tumor cells to evade and suppress anti-tumor immunity is a hallmark of cancer [3]. Tumor cells escape immune surveillance by downregulating antigen-presenting machinery and interferon (IFN) signaling pathways. Tumors also establish an immunosuppressive microenvironment by recruitment of regulatory T-cells (Tregs) and myeloid-derived suppressor cells (MDSCs), as well as the production of pro-tumor and anti-inflammatory agents such as transforming growth factor β (TGF-β), interleukin (IL)-10, and indoleamine 2,3-dioxygenase (IDO) [4].

Despite the immunosuppressive nature of most cancers, immune checkpoint inhibitors (ICIs) have emerged as an effective treatment in various solid tumors. However, only a fraction of cancer patients currently benefits from the long-lasting effects of ICIs. In contrast, most patients show primary resistance to these drugs. Furthermore, ICIs often induce immune-related adverse events that mimic autoimmune conditions in patients [5]. Therefore, elucidating the factors involved in the efficacy and toxicity of these agents is of significant value to those involved in clinical research and patient care.

The gut microbiota is a major metabolic organ in the human body that comprises diverse microorganisms, including bacteria, archaea, viruses, single-cell eukaryotes, and fungi. The human microbiome refers to the cluster of microbial genes in and on the human body. Most of the human microbiome resides in the gut, which can impact cancer development by direct and indirect interactions with tumor cells, the immune system, and the induction of metabolic inflammation [6]. Moreover, the gut microbiome has recently been suggested as a potential immune system modulator to fight against cancer progression [7]. In addition, microbiota-derived metabolites such as short-chain fatty acids (SCFAs) can modulate the tumor microenvironment (TME) and potentially enhance anti-tumor immunity. This, in turn, can improve the therapeutic response to ICIs [8]. Hence, in this review, we will highlight the known impacts of the gut microbiome and its metabolites on systemic and anti-tumor immunity and its impact on tumor progression and response to systemic therapy, including immunotherapy.

## 2. Microbiome and Immune System

The intestinal mucosal epithelium is comprised of intestinal epithelial cells (IECs), intraepithelial lymphocytes (IELs), paneth, and goblet cells as specialized secretory epithelial cells. Lamina propria is a connective tissue located beneath the epithelium mainly composed of Peyer’s patches consisting of different immune cells, including innate lymphoid cells (ILCs), inducible natural killer (iNK) cells, T and B lymphocytes as well as microfold cells (M cells). These cells are the communication bridge between the intestinal lumen and antigen-presenting cells [9,10,11]. In addition, Paneth cells and IECs strongly contribute to the host defense by secreting antimicrobial peptides (AMPs), such as α-/β-defensins and cathelicidins. Moreover, IECs and innate immune cells express pattern recognition receptors (PRRs), including toll-like receptors (TLR) and nod-like receptors (NLR) that recognize pathogen-associated molecular patterns (PAMPs) and damage-associated molecular patterns (DAMPs) [12].

Notably, Peyer’s patches are considered a well-organized germinal center composed of diverse B-cell repertoires characterized as a critical part of the humoral mucosal immunity by IgA antibody production [13]. Furthermore, lamina propria supports immunity toward the commensal bacteria in the gut lumen. It is mainly comprised of CD4^+^ T-cells, especially Foxp3^+^ Tregs and Th17 cells that exert immunomodulatory effects and maintain intestinal homeostasis [6]. In addition, CD8^+^ T-cells make up a critical population in direct contact with IECs in the small intestine. There are fewer CD8αβ^+^ T-cells than CD4αβ^+^ T-cells in the small and large intestinal epithelium [14]. Local CD8^+^ T-cells are fundamental in targeting intracellular pathogens and tumor cells. Also, CD8^+^ T-cells in the intestine, particularly IELs, contribute to tissue homeostasis and epithelial repair through the production of antimicrobial factors and tissue repair factors in response to intestinal microbiota [15].

Metagenomic gene sequencing has uncovered about 3.3 million non-redundant microbial genes in individuals, approximately one hundred-fold more than the host. Additionally, over 99% of the genes in human fecal samples are bacterial genes [16]. Therefore, the human microbiome is a wide-broad invisible organ that can be considered the second genome of the body. The gut microbiota, where most microbial genes are present in the human body, is essential in different physiological functions such as metabolism, immune regulation, and homeostasis [17]. It has been demonstrated that the gut microbiome promotes the development and differentiation of innate and adaptive immune cells, particularly Th1, Th17, and Tregs [18].

Germ-free animals are valuable for studying the relationship between commensal microbiota and the host immune cells. Experiments in these gnotobiotic animals have confirmed microbiota-based immune development and immune homeostasis in early life [8,19]. The absence of the gut microbiota in germ-free mice is highly associated with immune dysfunction, including lymphoid tissue defects, smaller Peyer’s patches, reduced number of IELs, and an inadequate humoral mucosal immunity and IgA secretion, as well as a decreased number of immune cells, including Th1 and Th17 cells [20,21,22,23,24,25]. Moreover, Foxp3^+^ Tregs are significantly reduced in antibiotic-treated and germ-free mice, indicating the crucial role of the gut microbiota in Tregs development [26,27]. Instead, immune maturation occurs following microbiota transplantation to germ-free animals. This confirms the fundamental role of intestinal microbiota in immune regulation by the development and function of lymphoid cells, primarily via Tregs differentiation, which is essential for controlling inflammation and tissue homeostasis [6].

The development of Tregs is mediated by two important anti-inflammatory cytokines, IL-10 and TGF-β. The TGF-β-rich environment promotes the accumulation of Tregs in the tissues, and IL-10 makes a positive feedback loop for Treg differentiation [28,29]. Different bacterial species, such as *Clostridium* spp. and *Bacteroides fragilis,* facilitate the induction of Tregs [30]. Polysaccharide A (PSA) produced by *Bacteroides fragilis* contributes to Treg differentiation in a TLR2-dependent manner [31]. TLR2/TLR1 heterodimer and the Dectin-1 signaling pathway in response to *Bacteroides fragilis*-derived PSA activates the PI3K downstream pathway in antigen-presenting cells that could mediate the expression of anti-inflammatory agents such as IL-10 by T-cells [32]. In addition, *Bacteroides fragilis* supports Tregs in producing anti-inflammatory cytokine IL-10, which regulates intestinal homeostasis. *Bacteroides fragilis*-derived PSA enhanced Th1-related immune response and maturation of the host immunity [33]. Conversely, PSA derived from *Bacteroides fragilis* restrains Th17 development [31]. Evidence has suggested an association between decreased *Bacteroidetes, Lachnospiraceae, Roseburia hominis,* and *Faecalibacterium prausnitzii* with Th17-mediated diseases such as Crohn’s disease or ulcerative colitis due to Th17/Treg imbalance [34,35].

Microbiome-derived metabolites, including SCFAs, aryl hydrocarbon receptor (AhR), and TLR and NLR ligands, can influence the immune system locally and systemically. Anaerobic commensal bacteria such as *Bacteroidetes* and *Firmicutes* phylum produce SCFAs, including acetate, propionate, and butyrate, through dietary fiber fermentation. Butyrate and propionate produced by fiber-fermenting commensal microbes are linked to the upregulation of Foxp3^+^ Tregs development by inhibiting histone deacetylases (HDAC) inside the cell [36]. *Clostridium*-derived SCFAs can activate the latent form of TGF-β that acts as a potent inducer of Tregs [37]. It has been indicated that SCFAs mediate the activation of the signal transducer and activator of transcription 3 (STAT3) and the mammalian target of rapamycin (mTOR), which leads to the upregulation of transcription factor B lymphocyte-induced maturation protein 1 (Blimp-1). This pathway leads to Th1 development and SCFAs-mediated IL-10 production [38]. It has also been demonstrated that microbiota-derived metabolites potentially promote IFN-γ and T-bet expression, which results in Th1 development [39]. Additionally, butyrate could inhibit HDAC and the development of Th1 cells independent of the SCFAs receptor, GPR43, which is commonly expressed on a wide variety of immune cells, including neutrophils, macrophages, dendritic cells, mast cells, and lymphocytes [40,41]. Furthermore, butyrate potentially exerts its immunomodulatory effects through the induction of thymic and peripheral Tregs differentiation, inhibition of the nuclear factor kappa B (NF-κB) signaling pathway, and inducing several anti-inflammatory genes in dendritic cells [42].

A recent study demonstrated an improvement of Th1/Th2 and Th17/Treg imbalance following fecal microbiota transplantation (FMT) in an ulcerative colitis mouse model, indicating a role of the gut microbiota on T-cell differentiation and immune homeostasis [43].

Several metabolites, such as HpmA hemolysin and TcdA and TcdB toxins, are commonly produced by the intestinal resident microbes, including *Proteus mirabilis* and *Clostridium difficile.* These metabolites activate NOD-, LRR-, and pyrin domain-containing 3 (NLRP3), an intracellular sensor mainly expressed by gut epithelial cells and macrophages that detect microbial motifs [44]. Also, other microbiota-associated metabolites such as taurine, histamine, and spermine modulate the activation of the NLRP6 inflammasome, expressed by intestinal epithelial cells [45]. NLRP3 and NLRP6 inflammasomes mediate the activation of downstream pathways and pro-inflammatory cytokine production, such as IL-1β and IL-18. This immunological mechanism generally results in mucosal stability and sustained AMPs production [46]. Also, *Enterococcus faecium*-derived hydrolase and SagA induce AMP production by activating MyD88 and NOD2-mediated innate immune pathways [47]. The gut microbiota can also induce myeloid cells, functional ILCs, and IL-9-secreting T-cell populations in colonized hosts compared to germ-free and antibiotics-treated mice with impaired Th9 development. *Clostridia*-related segmented filamentous bacteria (SFB) induce the production of IL-23 by antigen-presenting cells that activate ILC3 to initiate an IL-23R/IL-22 circuit. This process produces serum amyloid A from epithelium that promotes IL-17 production by Th17 cells [24,48]. SCFAs such as propionate enhance dendritic cell hematopoiesis by increasing the number of dendritic cells and macrophage precursors that impact intestinal immunity to control the growth of invading pathogens [49] (Figure 1).

On the other hand, the immune system could modulate the gut microbiota to prevent local inflammation and maintain tissue integrity. For instance, AMPs such as α-defensins and cathelicidin secreted by paneth cells and the intestinal innate immune system potentially avoid increased systemic microbial translocation and inflammation [6,50]. Moreover, it has been indicated that intestinal antigen-presenting cells possibly migrate to the thymus in early life to present gut microbial antigens to thymocytes. This could influence the thymic selection and development of the microbiota-specific T-cells. This observation postulates that early life exposure to commensal bacteria possibly shapes up microbiota-specific T-cell receptor (TCR) repertoire and is potentially associated with immune disorders like inflammatory bowel disease [51]. Moreover, the aging process is associated with changes in the composition of the gut microbiome. For example, some notable age-related changes in microbial communities include a decrease in *Bifidobacterium* and *Lachnospiraceae* and an increase in *Clostridiaceae* and *Enterobacteriaceae*, which could potentiate pathological status and disease development [52,53]. Importantly, these changes in microbiome composition should be considered in the context of a particular disease or pathological state. For example, *Enterobacteriaceae* is enriched in older adults and younger children [52]. Moreover, older adults with frailty and gut permeability have lower levels of *Prevotella*, *Roseburia, Faecalibacterium*, *Blautia*, and *Megamonas,* and enrichment in *Akkermansia*, *Parabacteroides*, and *Klebsiella* that is associated with higher IL-6 and HMGB1 levels. This is while gut microbiome diversity was not different between older adults with frailty and gut permeability from those in the control group [54]. Hence, the human microbial content and the microbiota-derived byproducts could also modulate the immune system.

## 3. Cancer—Microbiome—Immune Axis

The human body is a complex ecosystem of a brilliant collaboration between symbiotic microbes and host cells (Figure 2). Plenty of exogenous and endogenous stimuli can influence the abundance and function of different microbial species that colonize distinct ecological niches. Commensal bacteria contribute to human health and disease, such as cancer and other chronic conditions [55]. Microbiota-induced inflammation has a pro-tumorigenic activity that could orchestrate cancer progression. The gut microbiota and immune system interaction occur through direct and indirect mechanisms. Based on the concept of molecular mimicry, microbes are recognized directly as antigens. Alternatively, they can provide co-stimulatory signals by producing pro-inflammatory cytokines to induce immune responses. Microbial antigens sharing homology with tumor-specific epitopes could play an essential role in tumor immune surveillance [56]. Whole-exome sequencing has revealed neoantigens in melanoma and pancreatic cancer tumors homologous with microbial antigens [57,58]. For instance, an epitope found in *Bifidobacterium breve* represents a homology with neoantigen in a melanoma cell line. *Bifidobacterium breve* colonization in mice bearing melanoma tumor antigen results in specific T-cell response and decreased tumor growth [59]. It has also been indicated that the epitope of tail length tape measure protein (TMP) in enterococcal bacteriophage could mount a specific T-cell response representing cross-reactivity between tumor MHC class I–restricted antigens upon anti-PD-1 immunotherapy [60]. The cross-reactivity between microbial and tumoral antigens induces CD8^+^ T-cell response, which could regulate anti-tumor immunity and tumor response to immunotherapy [60]. Whole-exome sequencing and in silico neoantigen prediction have demonstrated that a high level of CD8^+^ T-cells specific to MUC16 neoantigens is associated with long-term survival in patients with pancreatic cancer. Also, specific T-cells that cross-recognized tumor antigens and microbial epitopes have been found in melanoma patients [60]. Balachandran VP et al. have also demonstrated that T-cells derived from peripheral blood of PDAC patients are most reactive to MUC16 neoantigens. However, they can also recognize microbial antigens. Therefore, higher neoantigen quality and potential cross-reactivity to microbial antigens indicate immunogenicity in tumors, which could increase the effectiveness of immunotherapies [58]. Thus, the gut microbiota could be considered an abundant reservoir of various antigens resembling tumor-specific epitopes. This phenomenon is critical in diseases such as melanoma and lung cancer, where immunotherapy is currently approved and commonly used.

Alternatively, microbial byproducts such as metabolites and toxins are considered a dynamic metabolic system that can orchestrate immunomodulation directly and indirectly. For instance, *Escherichia coli*-derived colibactin could exert a direct tumorigenic effect through adenylation of the DNA and induction of double-stranded DNA damage. This genetic mutation caused by direct exposure to pathogenic colibactin-producing bacteria has also been reported in colorectal cancer (CRC) patients [61]. It has been indicated that colibactin-positive *Escherichia coli* is associated with a reduction of CD3^+^CD8^+^ T-cells in a mouse model of CRC [62]. In addition, some bacterial toxins directly target neutrophils and macrophages by manipulating cell signaling and induction of cell death [63]. For example, Clostridial C3 toxins can interfere with macrophage function [64]. In addition, the immune dysregulation caused by other commensal microbial species such as *Streptococcus gallolyticus*, *Enterococcus faecalis*, *Bacteroides fragilis*, *and Fusobacterium nucleatum* also contribute to CRC development [65].

Furthermore, it has been demonstrated that gut microbiota such as *Escherichia coli* stimulates metastasis-related secretory protein cathepsin K, which is a crucial mediator between dysbiosis and tumor burden. Cathepsin K mediates M2 macrophage polarization through a TLR4-dependent pathway and supports tumor metastasis in CRC [66]. On the other hand, an in vitro study has shown that *Escherichia coli*-derived extracellular SCFAs, mainly acetic acid, elicit high cytotoxic effects on CRC and breast cancer cell lines [67]. Additionally, SCFAs, such as acetate, propionate, and butyrate, are shown to induce effector Th1 and Th17 functions in the kidney, which mediates ureteritis and hydronephrosis [68]. Ryu TY et al. have also demonstrated that microbiome-derived propionate could induce apoptosis in CRC cells by upregulation of tumor necrosis factor-alpha (TNF-α)-induced protein 1 (TNFAIP1) [69].

More evidence has unraveled bacterial determinants, such as gallic acid, lithocholic acid, and de-conjugated estrogens, which tremendously influence the mitochondrial dynamic and tumor progression [70,71,72]. *Fusobacterium nucleatum* may contribute to intestinal inflammation and promote CRC by secretion of outer membrane vesicles that could activate TLR4 and NF-κB signaling pathways. *Fusobacterium nucleatum* is highly enriched in tumor tissue samples from patients diagnosed with CRC. CRC is associated with the disruption of the colonic architecture with increased immune cell infiltration and depleted mucus layers [73,74]. Lactic acid bacteria (LAB) strains potently activate intracellular sensors, including stimulator of IFN genes (STING) and mitochondrial antiviral signaling (MAVS) that result in type I IFN (IFN-I) production, cytoprotective responses, and prevention of overreactive NF-κB-dependent inflammation in the gut [75]. Tikka C et al. demonstrated immune disruption and CRC progression following arsenic-induced dysbiosis. This is indirectly mediated by depletion of NOD2 and upregulation of inflammatory cytokines including TNF-α, IFN-γ, and IL-17, and reduction of anti-inflammatory mediators such as IL-10 [76]. It has been reported that the ratio of *Enterococcaceae* to *Bifidobacteriaceae* is significantly linked to gut damage and microbial translocation, which potentially promotes the development of hepatocellular carcinoma (HCC) through the production of pro-inflammatory cytokines and chemokines, including IL-6, IL-8, and monocyte chemoattractant protein-1 (MCP-1) as well as the downregulation of T-cell responses [77]. On the other hand, tumors potentially mediate aberrant activation of the c-Jun N-terminal kinase (JNK) pathway in intestinal tumor cells as well as enterocytes, resulting in an imbalance of gut microbiota, disruption of host-microbe homeostasis, and intestinal barrier dysfunction [78]. These are examples of pro-and anti-tumor effects of microbial products in the gut.

## 4. Gut Microbiota and Anti-Tumor Immunity

Innate and adaptive immune responses are vital components of the anti-tumor immunity against cancer. Diverse immune agents mediate tumor immune surveillance, but T-cell-mediated cytotoxicity is the principal mechanism of anti-tumor immunity. T-cells are central in anti-tumor response due to their ability to recognize specific peptides through the interaction of MHC-TCR, their cytotoxic effects, and their ability to exhibit immunological memory. Mainly, CD4^+^ and CD8^+^ T-cells are responsible for preventing tumor growth and cancer development. Therefore, ICIs mainly rely on T-cells for their efficacy. Notably, the gut microbiota is a tumor-extrinsic factor that can modulate anti-tumor defense mechanisms and impact the efficacy of cancer immunotherapy with ICIs (Figure 3).

The intestinal microbiota influences innate immune cells, including neutrophils, macrophages, NK cells, and γδ T-cells. Animal studies have shown that *Bifidobacterium longum 51A* could mediate the CXCL1 production and increase the accumulation of neutrophils. Moreover, oral treatment with *Bifidobacterium longum 51A* could enhance the myeloperoxidase activity of neutrophils and pro-inflammatory cytokine production, such as IL-6 and TNF-α [80,81]. Lakritz J et al. have shown that dietary administration of *Lactobacillus reuteri* is associated with reduced circulatory neutrophils and increased Foxp3^+^ Tregs [82]. It has also been investigated that the depletion of CD4^+^CD25^+^ Tregs is associated with mast cell accumulations in the mammary gland, increased mammary hyperplastic, and preneoplastic lesions in human epidermal growth factor receptor 2 (HER2) transgenic mice treated with *Lactobacillus reuteri* [82]. Furthermore, dietary administration *of Lactobacillus reuteri* in animals susceptible to breast cancer and on a high-fat diet decreased systemic inflammation and enhanced tumor inhibition [82].

Tumor-associated macrophages (TAM) are classified as anti-tumor M1 or pro-tumor M2 phenotypes. Microbiota perturbation following antibiotic treatment has been demonstrated to induce M2 macrophages, which partially promote tumorigenesis mediated by epithelial-mesenchymal transition (EMT) through TLR4/IL-10 signaling pathway [83,84]. Furthermore, *Escherichia*
*coli* colonization of the colon contributes to M2 macrophage polarization through TLR-4, which could promote tumor metastasis in CRC [66]. Also, M2 macrophages enhance the expression of PD-L1 in TME, potentially Reply to reviewer #suppressing the response to ICIs [85]. Moreover, tryptophan metabolism by intestinal microbiota induces the immunosuppressive phenotype of TAMs, most likely related to accelerated progression and high mortality rate in pancreatic ductal adenocarcinoma (PDAC) [86].

MDSCs are the hallmark of chronic inflammation in tissues. They are commonly present in tumors contributing to the immunosuppressive nature of TME. It has been shown that *Bacteroides fragilis*-derived IL-17 could potentially induce MDSCs in a Th17-dependent manner to promote colon tumorigenesis in MinApc^+/−^ mice [87]. NK cells are the principal innate immune arm involved in anti-tumor immunity through cytotoxic activity against tumor cells that escaped from CD8^+^ T-cell-mediated immunity by downregulating MHC-I [88]. The intestinal microbiota, such as *Fusobacterium nucleatum*, produce fatty-acid-binding protein 2 (Fap2) as an outer membrane protein that potentially binds to T-cell immunoglobulin and ITIM domain (TIGIT) receptors expressed on NK cells inhibiting the cytotoxic effects of these cells [89]. In addition, gut microbiota composition is positively associated with a high percentage of NK cells and a favorable response to ICIs in patients with non-small cell lung cancer (NSCLC) [90]. Hence, it is reasonable to presume that the disturbance of microbiota following antibiotic treatments may result in the reduced cytotoxic effect of NK cells and their tumor immune surveillance. For example, it has been shown that administration of azithromycin downregulates cytokine production and cytotoxic effects of NK cells, negatively impacting their anti-tumor functions [91].

The gut microbiota could also impact adaptive immune responses targeting tumor cells. *Bifidobacterium* spp. activate tumor-specific T-cells, increase the accumulation of CD8^+^ T-cells within melanoma and bladder tumors, and enhance IFN-γ production, which could slow down the growth of cancer cells by downregulating the NF-kB signaling pathway [92,93]. Moreover, *Bifidobacterium* strain enhances the efficacy of anti-tumor immune responses in colon cancer-bearing mice by increasing CD4^+^ and CD8^+^ T-cells, NK cells, and the CD4^+^/Treg, CD8^+^/Treg, and effector CD8^+^/Treg ratios [94]. In addition, the gut microbiota modulates the abundance of IFN-γ-producing CD8^+^ T-cells to influence colitis-associated tumorigenesis [95]. It has been observed that *Prevotellaceae* and *Anaeroplasmataceae* families are predictive of high and low tumor burdens of colon cancer, respectively [95]. Li Y et al. indicated that bacterial strains, especially members of *Bacteroides* and *Lactobacillus* are associated with improved anti-tumor immunity and higher infiltration of tumors by tumor-specific CD45^+^CD4^+^CD8^+^ T-cells and enhanced IFN-γ, TNF-α, and IL-2 production. This, in turn, could restrict melanoma growth in Rnf5^−/−^ mice [96]. In general, the gut microbiota and their metabolites influence anti-tumor immunity through different mechanisms, such as the development of Th1 and Th17 cells, induction of pro-inflammatory cytokines, and activation of MDSCs and NK cells. Microbiota-derived epitopes potentially stimulate antigen-presenting cells for further T-cell development and cytokine production, improving the systemic response to cancer immunotherapy. Considering the essential role of gut microbiota in balancing anti-tumor versus pro-tumor immune responses, it is vital to develop microbiome screening and therapeutic strategies that can help tip the balance in favor of anti-tumor immunity. Given the negative role of antibiotics on some critical components of innate and adaptive immunity, it is crucial to avoid broad-spectrum antibiotics in cancer patients receiving ICI therapy as much as possible. Further clinical studies are required to determine the potential benefit of microbiome supportive therapies such as FMT from healthy donors in cancer patients that require antibiotic treatment.

## 5. Gut Microbiota and Response to Systemic Therapy, including Immunotherapy

The interaction of immune checkpoint molecules such as PD-1 and its ligand, PD-L1, suppresses T-cell function and infiltration to the TME. The PD-1/PD-L1 interaction contributes to immune tolerance and, ultimately, immune escape of tumor cells [97]. The interaction of PD-1, mainly expressed on T-cells in the late phase of their activation, with its ligand PD-L1, expressed on tumor cells or other cells in the TME, inhibits the activation and effector functions of tumor-specific T-cells [98]. In contrast, CTLA-4 is expressed in the early phase of T-cell activation and competes with CD28 expressed on the surface of activated T-cells, with higher affinity, in binding to CD80 and CD86 on antigen-presenting cells preventing proper co-stimulatory signals for T-cell activation. Furthermore, Tregs constitutively express CTLA-4, allowing them to inhibit the activation of conventional T-cells [99]. Hence, ICIs, including anti-PD-1 (nivolumab and pembrolizumab), anti-PD-L1 (durvalumab, avelumab, and atezolizumab), and anti-CTLA-4 (ipilimumab and tremelimumab) antibodies have been developed for cancer immunotherapy [100]. Most patients remain unresponsive to ICIs despite the ability of these drugs to reinvigorate tumor-reactive T-cells in clinical settings [101]. Therefore, primary resistance to ICIs is an immense clinical problem that requires novel combination treatment strategies to improve the efficacy of these drugs.

Intratumoral microbes possibly mediate resistance to immunotherapy with ICIs and other forms of systemic therapy such as chemotherapy. In some cases, bacteria are found in patients’ tumors and genetically engineered mouse models of pancreatic cancer that are associated with a more immunosuppressive TME [102]. It can be postulated that targeting intratumoral microbes with antibiotics could modulate chemotherapy resistance due to the direct tumor-supportive roles of intratumoral bacteria, such as bacterial-induced autophagy in tumor cells [103]. Moreover, intratumoral bacteria possibly diminish the efficacy of systemic cancer therapy via metabolizing the chemotherapeutic drug to its inactive form [104]. Therefore, lower chemotherapy drug concentrations can be achieved due to the presence of bacteria in human tumors than in other organs [105].

Some intratumoral bacteria can negatively impact anti-tumor immunity. In contrast, others can potentially prevent cancer progression by providing bacterial antigens in the tumor that can mimic neoantigens and activate anti-tumor immunity. For example, RNA sequencing and immunopeptidomics analysis have recently identified 248 and 35 unique HLA-I and HLA-II peptides derived from 41 intratumoral bacterial species from 17 metastatic melanoma tumors. Microbial neoantigens in melanoma tumors are processed, presented, and recognized by T-cells [105]. These findings confirmed that cancerous tissue could present bacterial neoantigens to tumor-infiltrating T-cells and reinforce immune TME. Furthermore, the presence of microbes in the tumor can potentially improve dendritic cell maturation. Indeed, dendritic cells stimulated with live or heat-killed commensal bacteria can express co-stimulation/maturation markers and produce pro-inflammatory cytokine/chemokine, such as IL-1β and TNF-α [106]. Also, combining anti-PD-1 immunotherapy with bacterial therapy using *Clostridiales* strains cleared almost all tumor cells and reduced the volume and weight of melanoma tumors [107]. Therefore, clinical studies based on bacteria-based therapies in the form of complete or partial consortia are warranted to sensitize tumors to ICI therapy.

The resemblance of tumor-associated antigens and microbiota-derived epitopes potentially supports anti-tumor immunity. However, the intestinal microbiota has a dual effect on immunotherapy by enhancing or diminishing anti-tumor immune responses [108]. For instance, bacterial species including *Faecalibacterium*, *Bifidobacterium*, *Lactobacillus*, *Akkermansia muciniphila*, and *Ruminococcaceae* spp. play a considerable role in anti-tumor immune surveillance as well as the response to ICIs therapy [109,110]. Several seminal studies have already shown that intestinal microbiota composition is perturbed during cancer progression [111,112,113]. Microbiome sequencing and immune profiling of 233 patients with metastatic melanoma who received anti-PD-1 immunotherapy have shown that those with a more diverse gut microbiome had a higher ORR and improved survival outcomes. These findings indicate that a reduction in microbial diversity known as dysbiosis can result in poor response to ICIs [114]. Analysis of fecal microbiome signatures of 94 melanoma patients who received anti-PD-1 immunotherapy showed that *Ruminococcus (Mediterraneibacter) torques, Blautia producta, Blautia wexlerae, Blautia hansenii, Eubacterium rectale, Ruminococcus (Mediterraneibacter) gnavus,* and *Anaerostipes hadrus* are increased in non-progressors. In comparison, the stool samples of progressors are enriched with *Prevotella* spp., *Oscillibacter* spp., *Alistipes* spp., and *Sutterellaceae* spp. Moreover, transcriptomic analyses of fecal samples of those patients have identified a remarkable upregulation of superoxide dismutase (SOD2), pro-inflammatory cytokines such as IL-1β and CXCL8, transcription factors NFKBIZ, NFKBIA, TNFAIP3, and LITAF in progressors. Fecal samples of progressor also had an abundance of inflammatory cells, including dendritic cells, monocytes, macrophages, and neutrophils [115]. Furthermore, shotgun metagenomic sequencing of fecal samples from 165 non-resectable advanced (stage III or IV) cutaneous melanoma patients prior to immunotherapy with ICIs including nivolumab, pembrolizumab or ipilimumab, or a combination of ipilimumab and nivolumab revealed a significant association between the composition of the gut microbiome especially the special panel of *Bifidobacterium pseudocatenulatum*, *Roseburia* spp. and *Akkermansia muciniphila* with ORR and PFS [116].

The diversity of the gut microbiome composition is also correlated with the survival rates in response to chemoradiation in patients with cervical cancer [117]. In a prospective observational study on microbial composition in patients with metastatic renal cell carcinoma (mRCC) who had received nivolumab or nivolumab plus ipilimumab, it was demonstrated that the high diversity of gut microbiome profiles is strongly linked to the benefits in treatment outcomes [118]. In a randomized clinical trial of bacterial therapy combined with dual immunotherapy, fecal metagenomic sequencing of 30 mRCC patients with histology of clear cell and/or sarcomatoid and intermediate- or poor-risk disease demonstrated that ORR and PFS were significantly longer in patients who received nivolumab plus ipilimumab with CBM588 (a butyrate-producing strain of *Clostridium butyricum*) [119]. In addition, Jin Y et al. showed that in thirty-seven patients with metastatic, advanced stage IIIb/IV or recurrent NSCLC who were recruited from two clinical trials, CheckMate 078 [NCT02613507] and CheckMate 870 [NCT03195491] and treated with nivolumab there is a strong association between intestinal microbiome diversity and the responses to anti–PD-1 immunotherapy [90]. The studies mentioned above indicate that gut microbiome diversity is associated with better outcomes. However, prospective studies that would couple microbiome profiling in cancer patients receiving systemic or localized therapy with their clinical outcome can further determine whether diversity can be used as a predictive biomarker of response to systemic or localized therapy in various cancers. Furthermore, microbiome-modifying strategies, such as complete consortia FMT from healthy donors that can potentially increase gut microbiome diversity, can be used as supportive therapy for cancer patients receiving systemic or localized treatment.

A multicenter and retrospective study conducted by Takada K et al. has demonstrated that probiotics are linked to beneficial clinical outcomes in patients with advanced or recurrent NSCLC treated with anti-PD-1 monotherapy [120]. However, another study has reported paradoxical results in melanoma patients who took probiotic supplements while receiving immunotherapy and had a worse survival outcome. The obtained results are in line with the outcome of probiotic-treated mice, which have a lower frequency of IFN-γ-producing CD8^+^ T-cells in TME and impaired anti-tumor response compared to the controls [121]. A cohort study of 338 patients with NSCLC has demonstrated that intestinal *Akkermansia muciniphila* is significantly accompanied by clinical benefits with increased response rates and OS following PD-1 blockade [122]. Notably, the enhancement of bacterial compositions directly contributes to the efficacy of ICIs and improves clinical outcomes in cancer patients [123]. Modifying the gut microbiota in immunotherapy-refractory melanoma patients sensitized their tumors to anti-PD1 rechallenge [114]. FMT from patients with metastatic melanoma who had previously been treated with anti-PD-1 monotherapy and achieved complete response for at least one year to immunotherapy-refractory melanoma patients re-sensitized 30% of the treated patients to anti-PD1 treatment. FMT from patient donors modulated the immune cell infiltration and gene expression profiles in the TME [22]. Additionally, in another prospective study, FMT from long-term responder melanoma patients to anti-PD-1-refractory patients sensitized patients to anti-PD-1 rechallenge, further establishing the role of the gut microbiome in modulating response to immunotherapy [124]. In responding patients, the gut microbiota significantly shifted toward donor microbiota after FMT, and responders had decreased IL-8, IL-18, and CCL2 levels in their serum post-FMT [124]. Furthermore, FMT from long-term survivor patients with advanced PDAC by oral gavage to a mouse model of pancreatic cancer previously treated with antibiotics demonstrated active modification of the tumor microbiota with enriched *Clostridiales*, which inhibited tumor growth in an IFN-γ-producing CD8^+^ T-cell-dependent manner. In contrast, FMT from PDAC short term-survivors to mice resulted in an increased CD4^+^ Foxp3^+^ Tregs and MDSC infiltration [125].

It has been reported that intestinal microbiota composition influences the immunological complications of ICIs in solid tumors. For instance, Dubin K et al. have reported that microbiome composition in patients with metastatic melanoma who received ipilimumab treatment is significantly correlated with the development of immune-mediated colitis [126]. McCulloch JA et al. have reported that the abundance of *Lachnospiraceae* spp. and *Streptococcus* spp. are linked to the favorable clinical response and immune-related adverse events, respectively, upon anti-PD-1 treatment of melanoma patients [115]. It has also been demonstrated that increased bacterial species of *Bacteroidetes phylum* in the gut is significantly associated with resistance to the development of checkpoint-blockade-induced colitis [126]. Identifying the individual microbial species responsible for immunomodulation could open a new horizon toward personalized medicine in cancer immunotherapy. Hence, the microbiome has been identified as a robust predictive biomarker in response to immunotherapy, mainly the blockade of immune checkpoints. Intestinal *Bifidobacterium pseudolongum*-driven inosine metabolites promote Th1 differentiation and activation, shaping a robust immune response following anti-CTLA-4 and anti-PD-L1 immunotherapy in preclinical models of melanoma and CRC [127]. A preclinical study on mouse models of CRC has indicated that oral administration of *Clostridiales* strains actively leads to the intratumoral infiltration and activation of CD8^+^ T-cells. It has also been reported that the *Enterococcus* species secrete SagA enzyme leading to degradation of the bacterial cell wall, the release of muramyl peptide fragments, and activation of the NOD2 signaling pathway, which in turn improves the response to anti–PD-L1 immunotherapy in the mouse models of melanoma and colon cancer [128]. Based on these findings, it could be proposed that treatment with some bacterial strains as a limited consortium of probiotic therapy may improve the efficacy of anti-PD-1/PD-L1 therapy.

It has been proposed that cancer-mediated alterations in the gut microbiome could be detrimental to the efficacy of chemotherapy and/or ICIs in patients with NSCLC, renal cell carcinoma (RCC), and melanoma. At the same time, using antibiotics, probiotics, FMT, or nanotechnologies to modulate the gut microbiota can potentially reinforce anti-tumor effects of chemo drugs or ICIs [129,130]. Zackular JP et al. have reported that manipulating the gut microbiota using antibiotics could reduce the tumor burden in a mouse model of colon cancer. Moreover, early exposure to antibiotics significantly prevents tumorigenesis in a murine model of inflammation-driven CRC, which could be a helpful therapeutic approach in CRC management [131]. A retrospective study on 120 patients diagnosed with CRC indicated that antibiotics treatment two weeks prior to starting oxaliplatin-based therapy significantly improved the objective response rate (ORR) and disease control rate in advanced CRC. Also, the overall survival (OS) and progression-free survival (PFS) of CRC patients were significantly higher in the antibiotics-treated group [132]. The administration of antibiotic cocktails containing ampicillin, neomycin, metronidazole, and vancomycin significantly decreased the number and size of tumors, histological scores, and expression of pro-inflammatory cytokines such as IL-1β, IL-6, and IL-22. This reduced the tumorigenesis in a CRC mouse model [133]. On the other hand, antibiotic treatment negatively impacts the diversity of the gut microbiome, leading to altered metabolome and subsequent antibiotic resistance. The use of antibiotics affects the microbial composition and may reduce the efficacy of ICIs immunotherapy [134]. However, the negative effect of microbiome alterations with antibiotics is cancer and treatment-dependent. For example, a retrospective cohort study of 147 patients with metastatic colorectal cancer (mCRC) who have received bevacizumab (a VEGF inhibitor) has revealed that antibiotic use was inversely associated with a mortality rate [135]. A comprehensive literature analysis of 2740 patients with melanoma, NSCLC, RCC, and urothelial carcinoma from 19 eligible studies who were treated with ICIs, including anti-PD-1/PD-L1 inhibitors and/or anti-CTLA-4 inhibitors, has revealed that antibiotics use are inversely associated with OS and PFS in cancer patients [136]. The pooled results of 44 cohort studies on 12,492 cancer patients have revealed that antibiotic administration was significantly correlated with worse ORR, OS, and PFS [137]. A retrospective study on 234 patients with NSCLC who had received antibiotics, cephalosporin, and quinolones, within 60 days before the initiation of nivolumab and ipilimumab treatment exhibited shorter OS and PFS in comparison with patients who had not received antibiotics [138]. However, the international cohort study of 450 patients with HCC has indicated that antibiotic treatment 30 days before or after immunotherapy with ICIs, including anti-PD-1/PD-L1 inhibitors and/or anti-CTLA-4 inhibitors, has significant benefits in response to the treatment with prolonged PFS [139]. Chen L et al. reported that tumorigenesis is significantly reduced in ovarian cancer mouse models following the treatment with a cocktail of metronidazole, vancomycin, and streptomycin antibiotics [140]. However, increasing evidence shows that antibiotic therapy prior to ICIs treatment is detrimental in most settings where immunotherapy is approved. A cohort study of 137 men and 59 women with NSCLC, melanoma, and other tumor types revealed that β-lactam–based antibiotic therapy within 30 days prior to anti- PD-1/PD-L1 ICIs is significantly linked to worse OS and reduced response to treatment with ICIs [141]. The difference in the effect of antibiotics on patient outcomes seems to be directly relevant to the type of cancer. For example, in immune-hot tumors with a higher tumor mutation burden (TMB), such as NSCLC, the antibiotic effects seem to be detrimental to ICI efficacy. However, antibiotics seem to have a positive role in enhancing the effectiveness of systemic therapy in low TMB tumors with less immunogenic phenotypes such as CRC, HCC, and ovarian cancer.

Antibiotic-based treatments during the first six weeks of ICIs treatment have a maximum negative effect on response to the immunotherapy [142]. A systematic review and meta-analysis, which assessed forty-eight studies including 12,794 cancer patients, revealed that administration of antibiotics 30 days prior to ICIs treatment is negatively associated with OS and PFS [143]. Moreover, a retrospective cohort study on 228 patients with high-risk hematologic malignancies who have received in particular piperacillin/tazobactam, meropenem, and imipenem/cilastatin four weeks prior to the CD19 CAR T-cell therapy has demonstrated that the alteration of the intestinal microbiome is accompanied with reduced survival and increased immune effector cell-associated neurotoxicity syndrome. Also, stool sample profiling showed a low diversity of microbes with enrichment of *Akkermansia* in patients who received CD19 CAR T-cell therapy compared to the healthy volunteers. While the composition of the fecal microbiome with high abundances of commensal *Clostridia*, including the genera *Ruminococcus* and *Faecalibacterium*, family *Ruminococcaceae* and the species *Faecalibacterium prausnitzii* and *Ruminococcus bromii* were correlated with complete response to the treatment as well as no toxicity [144]. Microbiota-produced metabolites are one of the plausible mechanisms of immune maturation through enhanced memory responses of CD8^+^ T-cells. Butyrate and propionate are associated with increased oxidative phosphorylation and enhanced T-cell responsiveness to IL-15 [145,146]. A cohort study of 128 metastatic melanoma patients has indicated that sufficient dietary fiber intake is associated with beneficial outcomes of ICIs [121]. Microbial signatures are now being identified across all tumors based on histology and the cancer genome atlas (TCGA) database with opportunities to target them to improve outcomes and prevent cancer [147,148]. The gut microbiome is likely to become the newest frontier in oncology, and different diagnostic and therapeutic strategies involving the gut microbiome could be used in patient care in the near future. Several clinical trials are currently underway to determine the effect of microbiome-based strategies using FMT from healthy donors in combination with immunotherapy in immunotherapy-naïve patients with advanced melanoma (NCT03772899), RCC (NCT04163289), and lung cancer (NCT04951583).

## 6. Conclusions

According to the mounting evidence in clinical and preclinical settings, the gut microbiota and its metabolites play a dual role in tumor progression through pro-tumorigenic and anti-tumorigenic effects. While some bacterial species are linked to carcinogenesis, others modulate tumor growth and prevent cancer development. Although the exact mechanisms by which gut microbiota exerts the immunomodulatory effects are still unclear, the bulk of research hints that intestinal microbiota can be modified to improve clinical outcomes in various cancer settings. Given the close interaction between the immune system and gut microbiota and the beneficial impact of gut microbiota on immune development, bacterial-based therapies could be considered a promising strategy to improve the clinical outcome of cancer immunotherapy and other systemic therapies. Furthermore, microbiome screening strategies can potentially inform treatment strategies based on the presence or absence of specific organisms in the gut before the start of treatment. Microbiota is a modifiable organ that allows personalized cancer treatment strategies based on the type of cancer, treatment modality, and patients’ microbiome and immune profile in the near future.

Nevertheless, currently, there is no clear understanding regarding the function of each bacterium strain, mechanism of action in anti-tumor immunity, and therapeutic response in cancer treatment. Additionally, we do not fully understand the best microbiome profile from potential donors or the best-limited consortia makeup that would allow for the best therapeutic response in patients. These are significant limitations of this fast-growing field in oncology. However, developing bacteria-based therapies in oncology could open a new horizon in cancer treatment.

## Figures and Tables

**Figure 1 cancers-14-03563-f001:**
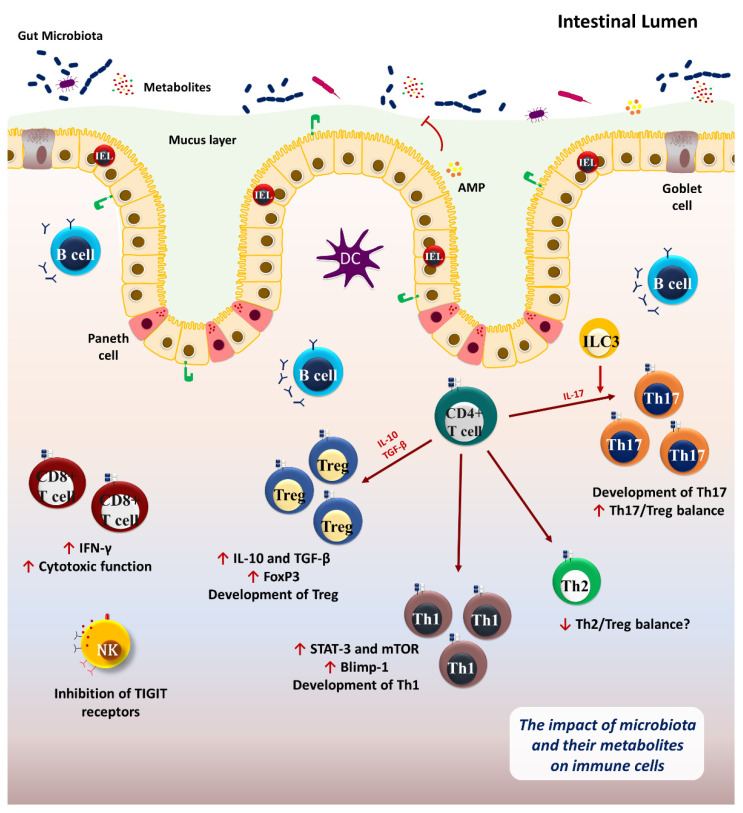
The interaction between gut microbiota and the immune system. The intestinal microbiota and their metabolites, including SCFAs, induce T-cell differentiation mainly on Tregs, Th1, and Th17 cells, impacting systemic immunity. The gut microbiota orchestrates Th1 differentiation from CD4^+^ T-cell by induction of STAT-3 and mTOR pathway. In addition, microbiota induces Treg development by the production of IL-10 and TGF-β. Microbiota and their metabolites may increase the Th17/Treg balance. The intestinal microbiota could activate CD8^+^ T-cells and NK cells which are critical in anti-tumor immunity. On the other hand, the immune system modulates systemic microbial translocation and inflammation through the activation of PRR signaling pathways as well as the production of AMPs. IEL: intraepithelial lymphocytes; PRR: pattern recognition receptor; AMP: antimicrobial peptides; DC: dendritic cell; NK: natural killer cell; ILC3: Type 3 innate lymphoid cells; TIGIT: T-cell immunoglobulin and ITIM domain.

**Figure 2 cancers-14-03563-f002:**
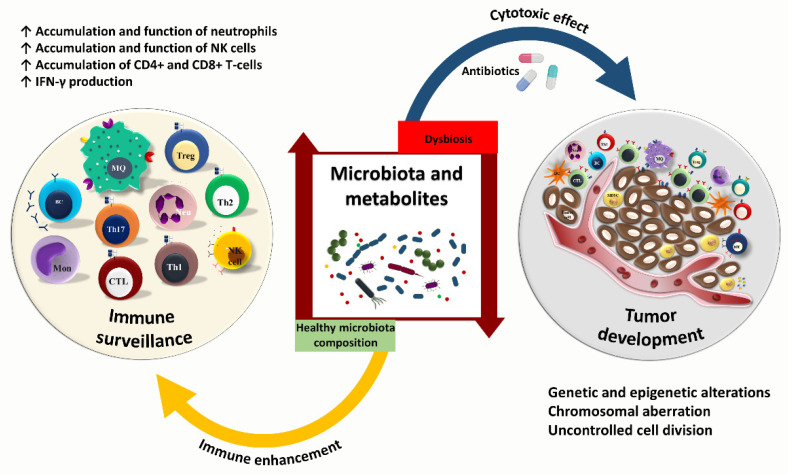
The cancer—microbiota—immune system axis. The gut microbiota plays a critical role in the immune maturation and prevention of cancer development. The increase in commensal microbiota results in the enhancement of the immune system. While the reduction of beneficial bacteria is linked to tumor progression. Healthy microbiota composition potentially increases the accumulation and function of neutrophils, NK cells, CD4^+^, and CD8^+^ T-cells. On the other hand, using antibiotics may lead to dysbiosis, which can negatively affect T-cell function. The perturbation of gut microbiota consequently results in tumor growth and cancer development.

**Figure 3 cancers-14-03563-f003:**
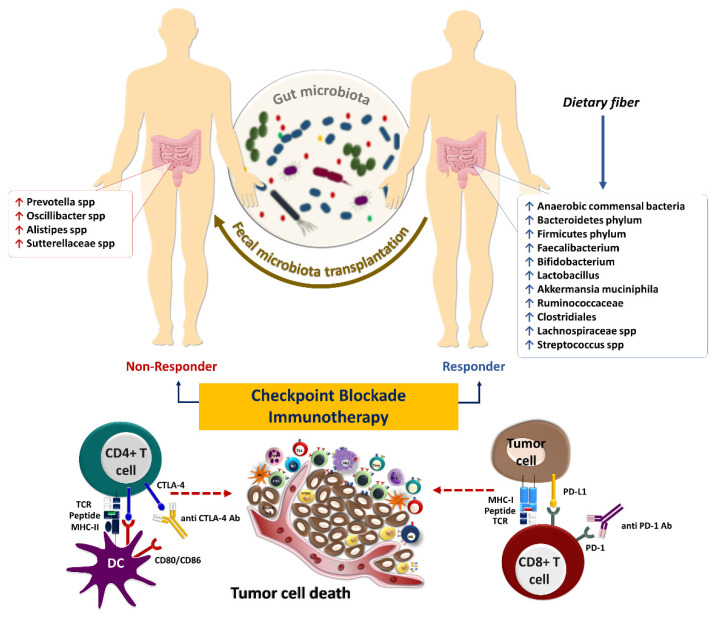
The influence of gut microbiota on the response to immune checkpoint inhibitors in cancer immunotherapy. Some microbes, including anaerobic commensal bacteria, which increase following sufficient dietary fiber intake, are associated with beneficial outcomes to checkpoint blockade immunotherapies such as anti-PD-1/PD-L1 and anti-CTLA4 antibodies [79]. Fecal microbiota transplantation could potentially improve the response in patients who failed to respond to such therapies.

## Data Availability

All the data covered in this manuscript is present in the paper.

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
