# Peer review of "The Effect of the Gut Microbiota on Systemic and Anti-Tumor Immunity and Response to Systemic Therapy against Cancer"

_cancers, 2022, doi:10.3390/cancers14153563_

Round 1

Reviewer 1 Report

Summary

This review focuses on the intricate relationship between microbiome, especially gut microbiota, and tumor-related immunity. It covers topics ranging from preclinical studies, focusing on bacteria-leukocyte signalings and their implication on tumor/host immunity, to clinical trials examining the correlation between microbiota composition and efficacy of checkpoint inhibitor immunotherapies. The authors compiled past and recent research findings stepwise to provide readers with a structured scheme on how different bacterial species can influence cancer survival outcomes by altering innate and adaptive immune systems. Overall, this review informs the critical role of microbiota on anti-tumor immunity. However, the manuscript covers topics and themes that were already extensively reviewed and published by others within the past 2 years [1-8]. Even for the updated research findings it still falls short to conclude novel patterns or to infer new directions of future research. On the other hand, this manuscript has many improper scientific nomenclature, improper reference statements, loose paragraph and sentence structures, misspelled words and misplaced comments. Therefore, it needs to check carefully its content and writing.

Specific comments:

11.     Improper scientific nomenclature: the scientific nomenclature of bacterial species should follow the International Code of Nomenclature of Prokaryotes, with both generic and specific names italicized (e.g., Escherichia coli, E. coli, Escherichia). Please rephrase all the scientific names in the manuscript.

22.   Statements that need clarifications

a.     Line 85-89: in the referenced article [5], I could not find any description stating CD8+ T cells are a smaller population than CD4+ T cells in lamina propria of the intestinal epithelium. Please indicate the reference to this statement or consider rewriting the populational relationship between the two types of T cells in the intestinal epithelium.

b.     Line 90-92: in the referenced article [11], I could not find the descriptions stating “(microbial genes) containing 99% of inherently modifiable genes in the human body”. Some close mentions include “Over 99% of the genes (in the faecal samples) are bacterial, indicating that the entire cohort harbours between …” and “It has been estimated that the microbes in our bodies collectively make up to 100 trillion cells … they encode 100-fold more unique genes than our own genome”. Please indicate the source of this statement or consider rewriting it.

c.     Line 100-102: in the referenced articles [14-16], I could not find any research findings suggesting decreased functions of Th1 and Th17 cells upon the absence of gut microbiota in germ-free mice. The only approximation is the increase of αβ T cells after microbial association in reference [16]. Please indicate the source of this statement (perhaps reference [5]) or consider rewriting it.

d.     Line 130-131: there is no reference for this following statement “Clostridium-derived SCFAs can activate the latent form of TGF-β that acts as a potent inducer of Tregs”. Please indicate a proper reference to this statement.

e.     Line 146-149: I could not find any statement supporting the role of NLPR3 in microbiota metabolites-mediated AMP production according to the referenced article [29]. In the referenced article [29], the authors even stated that “Consequently, taurine treatment also induced upregulation of AMPs. This induction of IL-18 was dependent on NLRP6, but not on NLRP3”. Please indicate a proper reference to this statement or consider rewriting it. Also, the authors clearly point out which metabolites they have incorporated in the study. Consider discussing them to improve readers’ understandings on the mechanism of microbiota metabolites-mediated AMP production.

f.      Line 152-153: According to the referenced article [32], “serum amyloid A (SAA) acted on lamina propria dendritic cells (LP DCs) to promote Th17 cell differentiation in vitro”. There is no mention of “IL3 cytokines production” in this article. Please delineate which IL3-related cytokines (IL-17, IL-22, GM-CSF etc.) are involved in development of Th17, or indicate a proper reference for this statement and consider rewriting it.

g.     Line 154-156: Please consider rephrasing this sentence “The recognition of SCFAs by host cells could also enhance the number of myeloid precursors such as dendritic cells that impact intestinal immunity to control the growth of invading pathogens” to clearly reflect the original statement in reference [33]: “Whereas SCFAs did not affect the generation of BM monocytes and neutrophils, they influenced DC hematopoiesis by increasing the number of macrophage and DC precursors and CDPs in the BM”.

h.     Line 158-160: Please consider rewriting or reciting proper references for this sentence “For instance, AMPs such as α-defensins and cathelicidin secreted by paneth cells and the intestinal innate immune system potentially avoid increased systemic microbial translocation and inflammation”. In one of the reference [34], the authors observed that α-defensin (HD-5) secreted by Paneth cells changes the microbial composition by its plethora of degraded fragments, but they did not discuss whether HD-5 (or HD-6 that is also a subject of their study) affect systemic microbial translocation or inflammation. Besides, in an earlier study by Jan Wehkamp et al. (Proc Natl Acad Sci U S A. 2005 Dec 13;102(50)), they have discovered that decreased α-defensin secreted by Paneth cells in patients with Crohn’s disease of ileum is not associated with the degree of inflammation. In the other reference [35], the authors only showed that cathelicidin can restore the death of ORAI1 knockout mice, yet they also did not prove that this restoration was due to cathelicidin attenuating microbial translocation or systemic inflammation.

i.      Line 166-169: Please consider rewriting or reciting proper references for this sentence “Moreover, the aging process is highly associated with decreased microbial diversity, highlighted by a decrease in Bifidobacterium and Lactobacillus and an increase in Enterobacteriaceae, which could strongly potentiate pathological status and disease development”. First, the decrease in Bactobacillus was not mentioned in the cited references [37, 38]. Second, the principal findings in [38], “Enterobacter and Klebsiella species were more frequently found in children while Proteus and Providencia species were typically found in the elderly. The species Bifidobacterium longum was the most frequently species isolated in children and adults, whereas Bifidobacterium adolescentis was the most encountered species in the elderly.”, were contradictory to [37]. Third, in [37], the authors “observed an increase in gut microbiota diversity with aging until the centenarian stage”, which was contradictory to the statement in the manuscript.

33.    Summarization on how bacterial composition influences cancer immunity and efficacy of immunotherapies can be improved

44.    Loose paragraph and sentence structures

55.    Please perform spell check again to rule out misspelled words and phrases.

66.    There seems to be a drafted comment on line 251-252. Please consider removing this sentence or rephrasing it.

References

[1] Pham, F., Moinard-Butot, F., Coutzac, C., & Chaput, N. (2021). Cancer and immunotherapy: a role for microbiota composition. European journal of cancer (Oxford, England : 1990)155, 145–154. https://doi.org/10.1016/j.ejca.2021.06.051

[2] Zhou, C. B., Zhou, Y. L., & Fang, J. Y. (2021). Gut Microbiota in Cancer Immune Response and Immunotherapy. Trends in cancer7(7), 647–660. https://doi.org/10.1016/j.trecan.2021.01.010

[3] Inamura K. (2020). Roles of microbiota in response to cancer immunotherapy. Seminars in cancer biology65, 164–175. https://doi.org/10.1016/j.semcancer.2019.12.026

[4] Araji, G., Maamari, J., Ahmad, F. A., Zareef, R., Chaftari, P., & Yeung, S. J. (2021). The Emerging Role of the Gut Microbiome in the Cancer Response to Immune Checkpoint Inhibitors: A Narrative Review. Journal of immunotherapy and precision oncology5(1), 13–25. https://doi.org/10.36401/JIPO-21-10

[5] Nomura M. (2022). Association of the gut microbiome with cancer immunotherapy. International journal of clinical oncology, 10.1007/s10147-022-02180-2. Advance online publication. https://doi.org/10.1007/s10147-022-02180-2

[6] Lu, Y., Yuan, X., Wang, M., He, Z., Li, H., Wang, J., & Li, Q. (2022). Gut microbiota influence immunotherapy responses: mechanisms and therapeutic strategies. Journal of hematology & oncology15(1), 47. https://doi.org/10.1186/s13045-022-01273-9

[7] Jain, T., Sharma, P., Are, A. C., Vickers, S. M., & Dudeja, V. (2021). New Insights Into the Cancer-Microbiome-Immune Axis: Decrypting a Decade of Discoveries. Frontiers in immunology12, 622064. https://doi.org/10.3389/fimmu.2021.622064

[8] Derosa, L., Routy, B., Desilets, A., Daillère, R., Terrisse, S., Kroemer, G., & Zitvogel, L. (2021). Microbiota-Centered Interventions: The Next Breakthrough in Immuno-Oncology?. Cancer discovery11(10), 2396–2412. https://doi.org/10.1158/2159-8290.CD-21-0236

Author Response

Reviewer #1

Comments and Suggestions for Authors

Summary

This review focuses on the intricate relationship between microbiome, especially gut microbiota, and tumor-related immunity. It covers topics ranging from preclinical studies, focusing on bacteria-leukocyte signalings and their implication on tumor/host immunity, to clinical trials examining the correlation between microbiota composition and efficacy of checkpoint inhibitor immunotherapies. The authors compiled past and recent research findings stepwise to provide readers with a structured scheme on how different bacterial species can influence cancer survival outcomes by altering innate and adaptive immune systems. Overall, this review informs the critical role of microbiota on anti-tumor immunity. However, the manuscript covers topics and themes that were already extensively reviewed and published by others within the past 2 years [1-8]. Even for the updated research findings it still falls short to conclude novel patterns or to infer new directions of future research. On the other hand, this manuscript has many improper scientific nomenclature, improper reference statements, loose paragraph and sentence structures, misspelled words and misplaced comments. Therefore, it needs to check carefully its content and writing. 

We thank the reviewer for their feedback. We have now fixed the errors that the reviewer has kindly pointed out and included more studies and more of our insight into the manuscript to differentiate the manuscript from those that have already been published.

Specific comments:

  1.    Improper scientific nomenclature: the scientific nomenclature of bacterial species should follow the International Code of Nomenclature of Prokaryotes, with both generic and specific names italicized (e.g., Escherichia coliE. coliEscherichia). Please rephrase all the scientific names in the manuscript.

We thank the reviewer for their helpful feedback. We have now fixed all the bacterial names in the text.

  1.  Statements that need clarifications
  2. Line 85-89: in the referenced article [5], I could not find any description stating CD8+ T cells are a smaller population than CD4+ T cells in lamina propria of the intestinal epithelium. Please indicate the reference to this statement or consider rewriting the populational relationship between the two types of T cells in the intestinal epithelium. 

We thank the reviewer for their feedback. We have modified the statement and have added a new reference (Line number: 90-96, Reference numbers: 14 and 15).

  1. Line 90-92: in the referenced article [11], I could not find the descriptions stating “(microbial genes) containing 99% of inherently modifiable genes in the human body”. Some close mentions include “Over 99% of the genes (in the faecal samples) are bacterial, indicating that the entire cohort harbours between …” and “It has been estimated that the microbes in our bodies collectively make up to 100 trillion cells … they encode 100-fold more unique genes than our own genome”. Please indicate the source of this statement or consider rewriting it. 

We thank the reviewer for their comment. We have re-written the statement to address the reviewer’s comment (Line number: 97-99).

  1. Line 100-102: in the referenced articles [14-16], I could not find any research findings suggesting decreased functions of Th1 and Th17 cells upon the absence of gut microbiota in germ-free mice. The only approximation is the increase of αβT cells after microbial association in reference [16]. Please indicate the source of this statement (perhaps reference [5]) or consider rewriting it. 

Thank you very much for your comment. New references have been added (Line number: 109-112, Reference number: 22, 23, 24, 25).

  1. Line 130-131: there is no reference for this following statement “Clostridium-derived SCFAs can activate the latent form of TGF-β that acts as a potent inducer of Tregs”. Please indicate a proper reference to this statement.

We thank the reviewer for their comment. A new reference has been added (Line number: 141-142, Reference number: 37).

  1. Line 146-149: I could not find any statement supporting the role of NLPR3 in microbiota metabolites-mediated AMP production according to the referenced article [29]. In the referenced article [29], the authors even stated that “Consequently, taurine treatment also induced upregulation of AMPs. This induction of IL-18 was dependent on NLRP6, but not on NLRP3”. Please indicate a proper reference to this statement or consider rewriting it. Also, the authors clearly point out which metabolites they have incorporated in the study. Consider discussing them to improve readers’ understandings on the mechanism of microbiota metabolites-mediated AMP production.

We thank the reviewer for their comment. We have now modified the statement and added more information on those metabolites to the text and added new references (Line number: 161- 171, Reference number: 44, 45, 47).

  1. Line 152-153: According to the referenced article [32], “serum amyloid A (SAA) acted on lamina propria dendritic cells (LP DCs) to promote Th17 cell differentiation in vitro”. There is no mention of “IL3 cytokines production” in this article. Please delineate which IL3-related cytokines (IL-17, IL-22, GM-CSF etc.) are involved in development of Th17 or indicate a proper reference for this statement and consider rewriting it.

We thank the reviewer for their comment. We have modified the sentence and included new information and added a new reference (Line number: 173-176, Reference number: 48).  

  1. Line 154-156: Please consider rephrasing this sentence “The recognition of SCFAs by host cells could also enhance the number of myeloid precursors such as dendritic cells that impact intestinal immunity to control the growth of invading pathogens” to clearly reflect the original statement in reference [33]: “Whereas SCFAs did not affect the generation of BM monocytes and neutrophils, they influenced DC hematopoiesis by increasing the number of macrophage and DC precursors and CDPs in the BM”. 

We thank the reviewer for their comment. We have rephrased the statement (Line number: 176-179)

  1. Line 158-160: Please consider rewriting or reciting proper references for this sentence “For instance, AMPs such as α-defensins and cathelicidin secreted by paneth cells and the intestinal innate immune system potentially avoid increased systemic microbial translocation and inflammation”. In one of the reference [34], the authors observed that α-defensin (HD-5) secreted by Paneth cells changes the microbial composition by its plethora of degraded fragments, but they did not discuss whether HD-5 (or HD-6 that is also a subject of their study) affect systemic microbial translocation or inflammation. Besides, in an earlier study by Jan Wehkamp et al. (Proc Natl Acad Sci U S A. 2005 Dec 13;102(50)), they have discovered that decreased α-defensin secreted by Paneth cells in patients with Crohn’s disease of ileum is not associated with the degree of inflammation. In the other reference [35], the authors only showed that cathelicidin can restore the death of ORAI1 knockout mice, yet they also did not prove that this restoration was due to cathelicidin attenuating microbial translocation or systemic inflammation. 

We thank the reviewer for their comment. Two new references have been added to the text (Line number: 181-183, Reference number: 6, 50). 

  1. Line 166-169: Please consider rewriting or reciting proper references for this sentence “Moreover, the aging process is highly associated with decreased microbial diversity, highlighted by a decrease in Bifidobacterium and Lactobacillus and an increase in Enterobacteriaceae, which could strongly potentiate pathological status and disease development”. First, the decrease in Bactobacilluswas not mentioned in the cited references [37, 38]. Second, the principal findings in [38], “Enterobacter and Klebsiellaspecies were more frequently found in children while Proteus and Providenciaspecies were typically found in the elderly. The species Bifidobacterium longum was the most frequently species isolated in children and adults, whereas Bifidobacterium adolescentis was the most encountered species in the elderly.”, were contradictory to [37]. Third, in [37], the authors “observed an increase in gut microbiota diversity with aging until the centenarian stage”, which was contradictory to the statement in the manuscript. 

We thank the reviewer for their helpful comment. We have modified the statement and added a new reference to better address the reviewer’s comment (Line number: 190-201, Reference number: 52, 53, 54). 

  1.   Summarization on how bacterial composition influences cancer immunity and efficacy of immunotherapies can be improved.

We thank the reviewer for their comment. The explanation regarding on how bacterial composition influences cancer immunity has been added to the text (Line number: 375-378).

  1.   Loose paragraph and sentence structures

We thank the reviewer for their helpful comment. We have revised the manuscript to address the reviewer’s comment.

  1.   Please perform spell check again to rule out misspelled words and phrases. 

We thank the reviewer for their feedback. We have spell-checked the manuscript and ruled out misspelled words.

  1.   There seems to be a drafted comment on lines 251-252. Please consider removing this sentence or rephrasing it.

We thank the reviewer for their helpful comment. We have now removed the sentence. 

References

[1] Pham, F., Moinard-Butot, F., Coutzac, C., & Chaput, N. (2021). Cancer and immunotherapy: a role for microbiota composition. European journal of cancer (Oxford, England : 1990)155, 145–154. https://doi.org/10.1016/j.ejca.2021.06.051

[2] Zhou, C. B., Zhou, Y. L., & Fang, J. Y. (2021). Gut Microbiota in Cancer Immune Response and Immunotherapy. Trends in cancer7(7), 647–660. https://doi.org/10.1016/j.trecan.2021.01.010

[3] Inamura K. (2020). Roles of microbiota in response to cancer immunotherapy. Seminars in cancer biology65, 164–175. https://doi.org/10.1016/j.semcancer.2019.12.026

[4] Araji, G., Maamari, J., Ahmad, F. A., Zareef, R., Chaftari, P., & Yeung, S. J. (2021). The Emerging Role of the Gut Microbiome in the Cancer Response to Immune Checkpoint Inhibitors: A Narrative Review. Journal of immunotherapy and precision oncology5(1), 13–25. https://doi.org/10.36401/JIPO-21-10

[5] Nomura M. (2022). Association of the gut microbiome with cancer immunotherapy. International journal of clinical oncology, 10.1007/s10147-022-02180-2. Advance online publication. https://doi.org/10.1007/s10147-022-02180-2

[6] Lu, Y., Yuan, X., Wang, M., He, Z., Li, H., Wang, J., & Li, Q. (2022). Gut microbiota influence immunotherapy responses: mechanisms and therapeutic strategies. Journal of hematology & oncology15(1), 47. https://doi.org/10.1186/s13045-022-01273-9

[7] Jain, T., Sharma, P., Are, A. C., Vickers, S. M., & Dudeja, V. (2021). New Insights Into the Cancer-Microbiome-Immune Axis: Decrypting a Decade of Discoveries. Frontiers in immunology12, 622064. https://doi.org/10.3389/fimmu.2021.622064

[8] Derosa, L., Routy, B., Desilets, A., Daillère, R., Terrisse, S., Kroemer, G., & Zitvogel, L. (2021). Microbiota-Centered Interventions: The Next Breakthrough in Immuno-Oncology?. Cancer discovery11(10), 2396–2412. https://doi.org/10.1158/2159-8290.CD-21-0236

Reviewer 2 Report

The manuscript was prepared very well. The introduction section justifies the purpose of the study. I congratulate the authors for the preparation of the manuscript

However, I have the following comments:

·       The tables/figures and the text describing them do not require any input, it is the strongest part of this study.

·       What is new in this manuscript for cancer patients?

·       Include a section on limitations and strengths.

·       What does this article contribute to, the authors should make their own assessment and include their own discussion of the results shown in the manuscript?

·       In the Conclusion section, state the most important outcome of your work. Do not simply summarize the points already made in the body — instead, interpret your findings at a higher level of abstraction. Show whether, or to what extent, you have succeeded in addressing the need stated in the Introduction (or objectives).

Author Response

Reviewer #2

Comments and Suggestions for Authors

The manuscript was prepared very well. The introduction section justifies the purpose of the study. I congratulate the authors for the preparation of the manuscript

However, I have the following comments:

  • The tables/figures and the text describing them do not require any input, it is the strongest part of this study.

We thank the reviewer for their encouraging feedback. We edited the figure legends to address the reviewer’s comment (Line number: 297-303).

  • What is new in this manuscript for cancer patients?

We thank the reviewer for their comment. We have organized the manuscript on logical progress from the interplay of the microbiome with systemic immunity, then microbiome and cancer, followed by the microbiome and anti-tumor immunity, and finally, the microbiome's effect on systemic therapy. In the last section “Gut Microbiota and Response to Immunotherapy, we discuss the potential clinical strategies to harness the power of bacterial-based therapies for improving treatment in cancer patients. We have also mentioned different clinical strategies and cited some of the ongoing clinical trials, including our ongoing trials in this space (Line number: 304- 387).

  • Include a section on limitations and strengths.

We thank the reviewer for their comment. We have added limitations and strengths to the conclusion section (Line number: 611-632). 

  • What does this article contribute to, the authors should make their own assessment and include their own discussion of the results shown in the manuscript?

We thank the reviewer for their comment. The new statements have been added to the conclusion section (Line number: 611-632). 

  • In the Conclusion section, state the most important outcome of your work. Do not simply summarize the points already made in the body — instead, interpret your findings at a higher level of abstraction. Show whether, or to what extent, you have succeeded in addressing the need stated in the Introduction (or objectives).

We thank the reviewer for their feedback. We have now added our insight from conducting microbiome intervention trials in cancer patients to various sections of the manuscript.

Reviewer 3 Report

General comments:

The authors have made an enormous effort to summarise available literature on the effects of the microbiota on immune maturation, tumor development, pro- and anti-tumor immune responses and the effects of chemo/immunotherapy. Figures are neatly designed, although not in all cases appropriate. The structure of many paragraphs is unclear and needs extensive revision in order to improve readability for the target audience. Rather than only mentioning, also discuss possible causes for discrepancies between results found in the literature.

-Title of the work does not match the content of the story: although the title suggests that the review is mainly focused on reciprocal interactions between the microbiota and the immune system in the context of anti-tumor immunity and immunotherapy, many of the cited work focusses on the role of the microbiota on cancer initiation, progression and/or responses to chemotherapy. Although effects of chemotherapy may partially be attributed to the immune system, this does not become apparent from the text. I would suggest to choose a more general title, or better, to leave out parts of the text that are not related to anti-tumor immunity/immunotherapy.

-Structure of the paragraphs: Most paragraphs begin with a clear sentence that introduces the topic of interest. However, on many occasions this introduction is followed by a citation or a statement that is seemingly unrelated to the topic. This significantly decreases the readability of the manuscript.

E.g. in lines 193-195 the principle of immunomodulation by bacterial toxins or byproducts is introduced. This is then directly followed by the “example” of E. coli-derived colibactin, but the genotoxic effect of colibactin in the cited study act on intestinal epithelial cells, and not on the immune system. Similarly, lines 204-206 refer to in vitro effects of SCFAs on CRC and breast cancer cell lines, not on immune modulation. Lines 202-203 include a statement on dysbiosis and tumor burden via cathepsin K. While this study does focus on effects on the immune system, this does not become clear from the text.

Another example is the sentence in lines 236-237, which states that the negative effects of antibiotic use may be cancer/treatment dependent. This is then followed by an enumeration of studies that show positive effects of antibiotic use. Also studies that investigate antibiotic use on cancer initiation are included, which seem unrelated to the topic. Much further in the manuscript, the negative effects of antibiotic use on immunotherapy are actually discussed.

Much of what is presented in paragraph 4 would also fit in 5. Perhaps 4 should only deal with effects of microbiota on immune cell populations and 5 only with interactions between microbiota composition and immune checkpoint blockade therapy.

Instead, paragraph 5, although entitled “Gut microbiota and Response to Immunotherapy”, also describes effects on chemotherapy (lines 358-364; lines 403-404).

-Use of abbreviations: Please avoid the use of unnecessary abbreviations of words that will not be re-used further in the text or are only used twice. E.g. ir-AEs in line 55 is forgotten when it comes back in line 442; DCR: line 245, CTSK: line 203, P-I-M: line 475, ICANS: line 478.  

-Details of cited studies: many of the sentences that describe a particular study include seemingly irrelevant information on the number of patients, the gender distribution of patients, or specific treatment details (e.g., check lines 413, 422, 453, 460 and more for relevance), while in other cases essential information on the cancer type, or disease model seems to be missing (e.g., line 229: in which cell type is the JNK pathway activated?, line 289: how was the microbiota perturbed?, line 320-322: in which cancer type?

-References:

Please include references for the statements made in lines:

-40-42
-87-89
-103-105
-135-137
-271-273
-282-285
-301-303
-348-351
-356-358
-358-360
-360-362
-380-382
-451-453

reference 87 refers to a review, but it would be more appropriate to refer to the original research article: Gut microbiome influences efficacy of PD-1–based immunotherapy against epithelial tumors; Routy et al., Science 2018

Other comments:

-CBI: the authors use the term ‘checkpoint blockade inhibitors’ (CBI) to describe anti-PD(L)1 and anti-CTLA-4 immunotherapies, which falsely suggests that these drugs inhibit a blockade. More appropriate would be to use for anti-PD(L)1 or anti CTLA-4 treatment the term ‘immune checkpoint blockade’ (ICB) or ‘immune checkpoint inhibitors’ (ICI) for anti-PD(L)1 or anti CTLA-4.

-Lines 157-158: I could somehow understand the statement that the interaction of the microbiota and the immune system is bi-directional. However, from the following it is unclear how this bi-directional interaction would operate in maintaining (epithelial) tissue integrity. Please explain this better.

-Lines 190-193: it would be helpful to cite here an example of microbial antigens resembling tumor-specific epitopes. And what would the consequence be? Reduced immunity of the tumor due to negative selection of tumor-specific T cells?? So please provide an example of this molecular mimicry and discuss the consequences for tumor immunity.

-Line 234: Please place a comma behind ‘melanoma’.

-Lines 251-251: This comment is correct, so please follow the comment and remove it.

-Lines 342-344: reformulate the sentence.

-Lines 464-465: This is a strong statement that does match with lines 470-473 but not with lines 454-456 and. In fact, rather than only mentioning results in lines 437-484, a profound discussion of their discrepancy is needed.

-Figure 1: Paneth cells are present outside of the crypt and even on top of the villus, which does not seem correct. Also, intestinal microbes are drawn in direct contact with the intestinal epithelium. However, direct contact between epithelial cells and the intestinal microbiota is avoided by the mucus layer, which is missing in this drawing. Finally, while the majority of the gut microbiota in humans resides within the colon, the drawing seems to represent the small intestine with a villus structure that is absent in the colon.

-Figure 2: The placement of the antibiotics in this figure suggests that antibiotic use directly promotes tumor development. However, the cited mouse model in line 240 and the other data cited till line 252 suggest the opposite. It would be more insightful to create separate panels that represent a ‘healthy’ diverse microbiota composition, and one that represents a state of dysbiosis.

Author Response

Reviewer #3

Comments and Suggestions for Authors

General comments:

The authors have made an enormous effort to summarise available literature on the effects of the microbiota on immune maturation, tumor development, pro- and anti-tumor immune responses and the effects of chemo/immunotherapy. Figures are neatly designed, although not in all cases appropriate. The structure of many paragraphs is unclear and needs extensive revision in order to improve readability for the target audience. Rather than only mentioning, also discuss possible causes for discrepancies between results found in the literature.

 We thank the reviewer for their encouraging feedback. We have modified the text and figures to address review's comments.

-Title of the work does not match the content of the story: although the title suggests that the review is mainly focused on reciprocal interactions between the microbiota and the immune system in the context of anti-tumor immunity and immunotherapy, many of the cited work focusses on the role of the microbiota on cancer initiation, progression and/or responses to chemotherapy. Although effects of chemotherapy may partially be attributed to the immune system, this does not become apparent from the text. I would suggest to choose a more general title, or better, to leave out parts of the text that are not related to anti-tumor immunity/immunotherapy.

We thank the reviewer for their helpful comment. We have now changed the title to a more general one, as the reviewer suggested.

-Structure of the paragraphs: Most paragraphs begin with a clear sentence that introduces the topic of interest. However, on many occasions this introduction is followed by a citation or a statement that is seemingly unrelated to the topic. This significantly decreases the readability of the manuscript.

We thank the reviewer for their feedback. We have now changed the main text's structure and added new paragraphs to address the reviewer's comments.

E.g. in lines 193-195 the principle of immunomodulation by bacterial toxins or byproducts is introduced. This is then directly followed by the “example” of E. coli-derived colibactin, but the genotoxic effect of colibactin in the cited study act on intestinal epithelial cells, and not on the immune system. Similarly, lines 204-206 refer to in vitro effects of SCFAs on CRC and breast cancer cell lines, not on immune modulation. Lines 202-203 include a statement on dysbiosis and tumor burden via cathepsin K. While this study does focus on effects on the immune system, this does not become clear from the text.

Another example is the sentence in lines 236-237, which states that the negative effects of antibiotic use may be cancer/treatment dependent. This is then followed by an enumeration of studies that show positive effects of antibiotic use. Also studies that investigate antibiotic use on cancer initiation are included, which seem unrelated to the topic. Much further in the manuscript, the negative effects of antibiotic use on immunotherapy are actually discussed.

We thank the reviewer for their feedback. We have now addressed the concern of reviewer (Line number: 253-255, Reference number: 62). 

Much of what is presented in paragraph 4 would also fit in 5. Perhaps 4 should only deal with effects of microbiota on immune cell populations and 5 only with interactions between microbiota composition and immune checkpoint blockade therapy.

Instead, paragraph 5, although entitled “Gut microbiota and Response to Immunotherapy”, also describes effects on chemotherapy (lines 358-364; lines 403-404).

We thank the reviewer for their feedback. We have now moved some statements from sections 3 and 4 to section 5. Also, section 5 was renamed “Gut Microbiota and Response to Systemic Therapy Including Immunotherapy” to accommodate the reviewer’s feedback.

-Use of abbreviations: Please avoid the use of unnecessary abbreviations of words that will not be re-used further in the text or are only used twice. E.g. ir-AEs in line 55 is forgotten when it comes back in line 442; DCR: line 245, CTSK: line 203, P-I-M: line 475, ICANS: line 478. 

We thank the reviewer for their feedback. The mentioned abbreviations have been removed across the text.

-Details of cited studies: many of the sentences that describe a particular study include seemingly irrelevant information on the number of patients, the gender distribution of patients, or specific treatment details (e.g., check lines 413, 422, 453, 460 and more for relevance), while in other cases essential information on the cancer type, or disease model seems to be missing (e.g., line 229: in which cell type is the JNK pathway activated?, line 289: how was the microbiota perturbed?, line 320-322: in which cancer type?

We thank the reviewer for their feedback. All references have been checked, and relevant information were included. For various clinical studies, we felt that including details such as the number of patients on the trial or the type of drug patients received would be necessary to the readers. However, that information is less crucial for preclinical studies. Please note, that there are new line numbers because of the edits in the document.

Line number: 473, reference number: 90

Line number: 491, reference number: 122

Line number: 564, reference number: 136

Line number: 572, reference number: 139

Line number: 290-293, information was added.

line 289 in the original file refers to the use of L. Reuteri as oral therapy for decreasing systemic inflammation in the study. The microbiome was perturbed by a high-fat diet. Information added to line 331

New line number: 364 tumor types were added

-References:

Please include references for the statements made in lines:

-40-42 (Reference number: 2)
-87-89 (Line number 90-96, Reference number: 14, 15)
-103-105 (Line number 112-114, Reference numbers: 26 and 27)
-135-137 (Line number 146-148, Reference number: 39)
-271-273 (Line number 317-319, Reference number: 79)
-282-285 (Line number 329-332: Reference number: 82)
-301-303 (Line number 349-352: Reference number: 88)
-348-351 (Line number 399-401: Reference number: 100)
-356-358 (Line number 407-409: Reference number: 102)
-358-360 (Line number 409-412: Reference number: 103)
-360-362 (Line number 412-414: Reference number: 104)
-380-382 (Line number 437-439: Reference number: 111, 112, 113)
-451-453 (Line number 556-557: Reference number: 134)

reference 87 refers to a review, but it would be more appropriate to refer to the original research article: Gut microbiome influences efficacy of PD-1–based immunotherapy against epithelial tumors; Routy et al., Science 2018

 We thank the reviewer for this valuable comment. We have set a reference for all statements (Line number: 437, Reference number: 110).

Other comments:

-CBI: the authors use the term ‘checkpoint blockade inhibitors’ (CBI) to describe anti-PD(L)1 and anti-CTLA-4 immunotherapies, which falsely suggests that these drugs inhibit a blockade. More appropriate would be to use for anti-PD(L)1 or anti CTLA-4 treatment the term ‘immune checkpoint blockade’ (ICB) or ‘immune checkpoint inhibitors’ (ICI) for anti-PD(L)1 or anti CTLA-4.

Thank you for your kind feedback. The word “CBIs” has been replaced by “ICIs” across the whole document.

-Lines 157-158: I could somehow understand the statement that the interaction of the microbiota and the immune system is bi-directional. However, from the following it is unclear how this bi-directional interaction would operate in maintaining (epithelial) tissue integrity. Please explain this better.

We thank the reviewer for their comment. We have modified the statement (line number 179)

-Lines 190-193: it would be helpful to cite here an example of microbial antigens resembling tumor-specific epitopes. And what would the consequence be? Reduced immunity of the tumor due to negative selection of tumor-specific T cells?? So please provide an example of this molecular mimicry and discuss the consequences for tumor immunity.

 We thank the reviewer for this valuable comment. The mentioned studies provided a list of peptides with microbial sources. We have referred to the studies so the readers can look at the list of peptides. We have also provided examples in the text. Please see lines 228, 231, and 240.

-Line 234: Please place a comma behind ‘melanoma’.

Applied (Line number: 539)!

-Lines 251-251: This comment is correct, so please follow the comment and remove it.

Thanks for the comment. The sentence has been removed!

-Lines 342-344: reformulate the sentence.

The sentence has been modified. New line 374

-Lines 464-465: This is a strong statement that does match with lines 470-473 but not with lines 454-456 and. In fact, rather than only mentioning results in lines 437-484, a profound discussion of their discrepancy is needed.

We thank the reviewer for this valuable comment. A new statement has been added to lines 583-589 to explain the discrepancy.

-Figure 1: Paneth cells are present outside of the crypt and even on top of the villus, which does not seem correct. Also, intestinal microbes are drawn in direct contact with the intestinal epithelium. However, direct contact between epithelial cells and the intestinal microbiota is avoided by the mucus layer, which is missing in this drawing. Finally, while the majority of the gut microbiota in humans resides within the colon, the drawing seems to represent the small intestine with a villus structure that is absent in the colon.

We thank the reviewer for their feedback. The figure has been modified to represent the microbial effect on the immune system in the colon.

-Figure 2: The placement of the antibiotics in this figure suggests that antibiotic use directly promotes tumor development. However, the cited mouse model in line 240 and the other data cited till line 252 suggest the opposite. It would be more insightful to create separate panels that represent a ‘healthy’ diverse microbiota composition, and one that represents a state of dysbiosis.

Thank you so much for your feedback.

The figure has been modified as the reviewer has suggested.

Round 2

Reviewer 3 Report

This manuscript has greatly improved and serves as an excellent introduction and reference to the microbiota literature that will certainly be welcomed and used by clinicians and scientist working in this expanding and highly relevant field. The authors have meticulously addressed the points raised. I have no further comments.